# Oxidative Modification, Structural Conformation, and Gel Properties of Pork Paste Protein Mediated by Oxygen Concentration in Modified Atmosphere Packaging

**DOI:** 10.3390/foods13030391

**Published:** 2024-01-25

**Authors:** Rui Liu, Wen Guan, Wei Lv, Zhuangli Kang, Qingling Wang, Duxin Jin, Xinxin Zhao, Qingfeng Ge, Mangang Wu, Hai Yu

**Affiliations:** 1College of Food Science and Engineering, Yangzhou University, Yangzhou 225127, China; ruiliu@yzu.edu.cn (R.L.); mz120211938@stu.yzu.edu.cn (W.G.); mz120211959@stu.yzu.edu.cn (W.L.); wangql891228@163.com (Q.W.); jinduxin@yzu.edu.cn (D.J.); xxzhao@yzu.edu.cn (X.Z.); qfge@yzu.edu.cn (Q.G.); mgwu@yzu.edu.cn (M.W.); 2School of Tourism and Cuisine, Engineering Technology Research Center of Yangzhou Prepared Cuisine, Yangzhou University, Yangzhou 225127, China; kzlnj1988@163.com

**Keywords:** pork paste, modified atmosphere packaging, protein oxidation, textural properties, water holding capacity

## Abstract

The objective of this study was to investigate the effect of pork oxidation through modified atmosphere packaging (MAP) on gel characteristics of myofibrillar proteins (MP) during the heat-induced gelation process. The pork *longissimus thoracis* (LT) was treated by MAP at varying oxygen concentrations (0, 20, 40, 60, and 80% O_2_) with a 5-day storage at 4 °C for the detection of MP oxidation and gel properties. The findings showed the rise of O_2_ concentration resulted in a significant increase of carbonyl content, disulfide bond, and particle size, and a decrease of sulfhydryl content and MP solubility (*p* < 0.05). The gel textural properties and water retention ability were significantly improved in MAP treatments of 0–60% O_2_ (*p* < 0.05), but deteriorated at 80% O_2_ level. As the concentration of O_2_ increased, there was a marked decrease in the α-helix content within the gel, accompanied by a simultaneous increase in β-sheet content (*p* < 0.05). Additionally, a judicious oxidation treatment (60% O_2_ in MAP) proved beneficial for crafting dense and uniform gel networks. Our data suggest that the oxidation treatment of pork mediated by O_2_ concentration in MAP is capable of reinforcing protein hydrophobic interaction and disulfide bond formation, thus contributing to the construction of superior gel structures and properties.

## 1. Introduction

Modified atmosphere packaging (MAP), also known as protective gas packaging, is a technology that involves extracting air from a package and injecting a pre-set gas mixture into the package [1,2]. Due to its ability to slow down natural product deterioration, MAP finds application in various product types, including meat, fish, fruits, and vegetables, to achieve preservation effects [3]. Usually, MAP employs a combination of two or three kinds of gases, typically carbon dioxide (CO_2_), oxygen (O_2_), and nitrogen (N_2_) [4]. In respect to MAP of raw meat, O_2_ is able to bind with surface myoglobin, forming oxygenated myoglobin, which increases the meat redness, and an increase of O_2_ concentration generally increases color stability [5]. However, MAP occupies ample space, limits the storage temperature, and the high content of O_2_ may lead to lipid and protein oxidation, thereby impacting meat quality [6].

Recently, the relationships between meat oxidation and the sensorial quality, physico-chemical properties, and nutritional characteristics in MAP have garnered considerable attention [7,8]. High-oxygen MAP often consists of 70–80% O_2_ and 20–30% CO_2_ for maintaining a vibrant red color in meat, but it also diminishes the edible quality of meat and affects its tenderness and juiciness [9,10,11,12]. Clausen et al. [13] demonstrated how high-oxygen MAP aggravated the oxidation of lipids and proteins and decreased protein degradation and meat tenderness of beef. Bao et al. [14] observed that as the O_2_ concentration increased (20–80%), protein oxidation levels elevated, subsequently leading to cross-link formation in structural proteins. Those cross-links, in turn, increased mechanical strength, ultimately resulting in tougher meat. Lund et al. [15] suggested that oxidation-induced myosin heavy chain cross-linking through disulfide bonding represents another mechanism contributing to the O_2_-induced toughening of meat. Many studies have thus focused on the influence of protein oxidation induced by O_2_ concentration in MAP on fresh meat quality. However, the effect on meat processing characteristics, especially the gel properties of the myofibrillar protein (MP), which are crucial functional attributes in the comminuted product matrix, has received little attention.

The mechanism of thermogelation of MP gel properties of meat has been extensively studied [16]. The MP undergoes denaturation and unfolding during heating, and the protein structure unfolds, exposing the functional groups that are originally in the protein [17]. The intermolecular interaction force is strengthened, the proteins aggregate with each other to form larger polymers, and finally, the proteins are rearranged into a structurally stable, viscoelastic three-dimensional gel network structure [18,19]. Disulfide bonds, in particular, play a significant role in protein cross-linking, contributing to enhanced gel hardness and water retention. Moderate oxidation-induced formation of disulfide bonds has been shown to be crucial for improving gel quality [20]. Wang et al. [21] investigated the impact of O_2_ concentration (0, 20, 40, 60, and 80%) in MAP on beef MP gel properties during heating. It was revealed that the 20% O_2_ treatment group produced a firmer and more elastic MP gel. However, the internal mechanism underlying the linkage of protein oxidation induced by MAP with gel properties warrants further investigation.

Up to the present, research concerning the oxidative modification of protein by O_2_ concentration in MAP have mostly emphasized clarification to the changes in raw meat quality. However, there are few studies on the effect of O_2_ concentration in MAP on the important functional properties of meat protein during the heat-induced gelation process. Consequently, this study aims to explore the impact of pork oxidation levels in situ through MAP with varying O_2_ concentrations (0, 20, 40, 60, and 80%) on gel characteristics of pork paste during the heat-induced gelation process. This work will enrich and develop the intrinsic relationship between the oxidative regulation of meat and protein gelation, thereby providing a theoretical basis for improving the quality of gel products.

## 2. Materials and Methods

### 2.1. Sample Preparation

#### 2.1.1. Raw Materials and Packaging

The pork was obtained from the 6-month-old crossbred pigs (Duroc × Landrace × Yorkshire) weighing 100 ± 10 kg (Sushi Meat Co., Ltd., Huai’an, Jiangsu, China). The slaughtering process was strictly implemented by the regulations of the Operating Procedures of Livestock and Poultry Slaughtering-Pig (GB/T 17236–2019) [22]. A total of 6 pork *longissimus thoracis* (LT) muscles from the right side of the pig carcasses were selected within 24 h post-slaughter. The pH was detected in the range of 5.5 to 5.8.

Visible connective tissue and external fat were meticulously removed from the surface of LT muscles. Subsequently, each muscle was sectioned into 13 chops, measuring approximately 100 g per chop (8 cm × 5 cm × 3 cm). The pH, weight, and color attributes of the 0 d sample were immediately measured. Three chops of each LT muscle were taken as 0 d biochemical meat samples for grinding and quick freezing with liquid nitrogen. Then, the other 10 chops from each LT sample were assigned to five MAP treatments with varying O_2_ concentrations (0, 20, 40, 60 and 80%). Thus, there were a total of 12 chops from 6 LT muscle (2 chop/LT) set in each MAP treatment. All packages were N_2_-balanced and contained 20% CO_2_ to inhibit microbial growth. Packaging was performed on a packaging machine X-500D (Shanghai Tan Xin Packaging Technology Co., Ltd., Shanghai, China) with a composite packaging bag (Dong Guang County Zhi Long Plastic Co., Ltd., Cangzhou, China) composed of polyamide/polypropylene. The O_2_ transmission rate was 12.707 cm^3^/m^2^/h/MPa. After the packaging was completed, the gas atmosphere (% O_2_, % N_2_, and % CO_2_) in package was verified by a multi-in-one gas detector (GT-903, Shenzhen Korno Electronic Technology Co., Ltd., Shenzhen, China), with no significant changes observed. All packages were placed in a refrigerator (Qingdao Haier Co., Ltd., Qingdao, China) for 5 days and the temperature was set to 4 °C, which was confirmed by a portable thermometer (ZG-8010, Ningbo Zhaoji Electrical Appliance Co., Ltd., Ningbo, China). After 5 days, the meat samples were ground and quick frozen with liquid nitrogen. Then, the samples were stored at −80 °C to enable further preparation of pork paste.

#### 2.1.2. Gel Preparation of Pork Paste

The pork paste was made according to the method of [23] with slight modification. The LT muscles were minced and then washed twice with deionized water (4 °C, 2 volumes, *w*/*v*), followed by one wash with 0.5% NaCl (4 °C, 2 volumes, *w*/*v*). The resulting paste was then centrifuged at 4000× *g* for 8 min to remove the supernatant. Subsequently, the paste was dissolved with 2.5% (*w*/*w*) NaCl. Protein concentration was measured by Biuret method and further adjusted to 100 mg/mL using distilled water. The gel was prepared by the process of heat-induced pork paste. Before heating, the paste was centrifuged at 500× *g* for 3 min to eliminate air bubbles. Afterward, the paste was put into a glass vial (5 cm height × 2.2 cm diameter), covered with aluminum foil, and then heated in an 80 °C water bath for 30 min. Finally, the gel was promptly chilled in ice water for 30 min and stored at 4 °C overnight.

### 2.2. Determination of Gel Quality

#### 2.2.1. Color

The color difference and whiteness of the gel was assessed following the approach outlined by Jeong et al. and Niu et al. [24,25], respectively. A colorimeter (CR-400, Konica Minolta, Japan) was used and calibrated with a calibration plate prior to each measurement. Measurements were taken at three different positions on each gel sample under the conditions of an illuminant D65, a 10° standard observer angle and a diameter of 8 mm. The average values of brightness (*L**), redness (*a**), and yellowness (*b**) were calculated. The color difference (Δ*E*) and whiteness were determined using Formula (1) and Formula (2), respectively.
(1)$$\Delta E=\sqrt{{\left({L_{1}}^{*}-{L_{2}}^{*}\right)}^{2}+{\left({a_{1}}^{*}-{a_{2}}^{*}\right)}^{2}+{\left({b_{1}}^{*}-{b_{2}}^{*}\right)}^{2}}$$
(2)$$\mathrm{Whiteness}=100\;-\sqrt{{\left(100\;-L^{*}\right)}^{2}+{a^{*}}^{2}+{b^{*}}^{2}}$$
where *L*_1_*, *a*_1_* and *b*_1_* are the values on day 0 sample and *L*_2_*, *a*_2_* and *b*_2_* are the values on MAP treatment samples.

#### 2.2.2. Texture Profile Analysis

After the gel was recovered to room temperature, textural properties were determined by a TMS-Pro texture analyzer (Food Technology Corporation, Sterling, VA, USA) [26]. The gel was shaped 20 mm in diameter and 23 mm in height with a mold of steel. The P/6 cylinder (diameter 12 mm) was selected as the test probe, the compression ratio was 50%, and extrusion took place twice. The test speed was set to 1 mm/s, while the pre-test and post-test speeds were both 5 mm/s. Based on the obtained force-time curve, the TPA attributes including hardness, cohesiveness, springiness, gumminess, and chewiness were recorded.

#### 2.2.3. Water Holding Capacity

The determination of water holding capacity is based on a previously reported method [25,27]. The pork paste was weighed before heating and recorded as W_0_. After cooking, the gel was extracted from the glass tube and gently dried using filter paper and was weighted as W_1_. The gel was then subjected to centrifugation at 10,000× *g* for 10 min at 4 °C. Afterward, the exudate was poured and wiped dry with absorbent paper and weighed as W_2_. Cooking loss and centrifugation loss were calculated according to Formula (3) and Formula (4), respectively.
(3)$$\mathrm{Cooking}\;\mathrm{loss}\;\left(\%\right)=\frac{W_{0}\;-\;W_{1}}{W_{0}}\;\times \;100\%$$
(4)$$\mathrm{Centrifugation}\;\mathrm{loss}\left(\%\right)=\frac{W_{1}\;-\;W_{2}}{W_{1}}\times 100\%$$

### 2.3. Determination of Protein Oxidation of Pork Paste

#### 2.3.1. Protein Surface Hydrophobicity

The determination of surface hydrophobicity was conducted following the method described by Chelh et al. [28] and slightly modified. A protein suspension with a concentration of 2 mg/mL was prepared, and 200 µL of 1 mg/mL bromophenol blue (BPB) was added to 1 mL of the protein suspension. After being fully mixed, the suspension was placed on a shaker and slowly shaken for 10 min. Then, the mixture was centrifuged at 4000× *g* for 15 min. The absorbance of the supernatant was measured at 532 nm, recorded as A_1_. As for the control, the protein suspension was replaced with 20 mM PBS, and the absorbance value was recorded as A_0_. The calculation formula was as follows:(5)$$\mathrm{BPB}\;\left(\mu g\right)=\frac{A_{0}-A_{1}}{A_{0}}\times 200\;\mu g$$

#### 2.3.2. Protein Solubility

According to the method developed by Wang et al. [29], 1 mL of 2 mg/mL protein suspension was placed at 4 °C for 1 h, and then centrifuged at 8000× *g* for 15 min. The resulting supernatant was collected to determine protein concentration using the Biuret method. The protein concentration was denoted as C_b_ before centrifugation and as C_a_ after the centrifugation process. The calculation formula was as follows:(6)$$\mathrm{Protein}\;\mathrm{solubility}=\frac{C_{a}}{C_{b}}\;\times 100\%$$

#### 2.3.3. Sulfhydryl and Disulfide Bond (S-S) Content

The determination of sulfhydryl groups and S-S groups of protein in pork paste was slightly modified based on the method outlined by Cui et al. [30]. Briefly, protein aliquots in buffer A (Tris-Gly) and buffer B (Tris-Gly-8M Urea) were treated with 5,5′-dithiobis-(2-nitrobenzoic acid) (DTNB) at 40 °C for 25 min. The absorbance of two samples were detected and recorded as S_A_ and S_B_. The free and total sulfhydryl groups were calculated using Formula (7) and Formula (8), respectively. As for protein S-S content, another aliquot in buffer C (Tris-Gly-10M Urea) with β-mercaptoethanol (β-ME) was created at 25 °C for 1 h, followed by trichloroacetic acid (TCA) treatment. After centrifugation, the pellet was re-dissolved in buffer B and DTNB was added. The absorbance was recorded as S_C_. The protein disulfide bond was calculated using Formula (9).
(7)$$F\mathrm{ree}\;\mathrm{sulfhydryl}\;\mathrm{content}\;(\frac{\mathrm{nmol}}{\mathrm{mg}})=73.53S_{A}\;\times \;D/C$$
(8)$$\mathrm{Total}\;\mathrm{sulfhydryl}\;\mathrm{content}\;(\frac{\mathrm{nmol}}{\mathrm{mg}})=73.53S_{B}\times \;D/C$$
(9)$$S\text{-}S\;\mathrm{content}\;(\frac{\mathrm{nmol}}{\mathrm{mg}})=73.53S_{C}\;\times \;D/C\;-\;\mathrm{total}\;\mathrm{sulfhydryl}\;\mathrm{content}$$
where D is the dilution coefficient and C (mg/mL) is the protein concentration in the tested sample.

#### 2.3.4. Dityrosine Content

The content of dityrosine was assayed following the method developed by Davies et al. [31] with slight adjustments. Protein suspension at 2 mg/mL was mixed with 20 mM PBS buffer containing 0.6 M of sodium chloride, and the mixture was centrifuged (5000× *g*, 10 min, 4 °C). The intensity of the fluorescence of the supernatant was detected with an excitation wavelength of 325 nm and an emission wavelength of 425 nm. Dityrosine content in the protein was obtained by dividing the fluorescence intensity by protein concentration.

#### 2.3.5. Carbonyl Content

The protein carbonyl of pork paste was measured by the derivatization of 2,4-dinitrophenylhydrazones (DNPH), which was described by Xu et al. [32]. The absorbance was recorded at 370 nm and protein content was assessed with the Biuret assay. The calculation formula is as follows:(10)$$\mathrm{Carbonyl}\;\mathrm{content}\;\left(\frac{\mathrm{nmol}}{\mathrm{mg}}\right)=10^{6}\times \;\frac{A}{C\;\times \varepsilon \times b}$$
where A indicates the absorbance of the sample, b is the optical path (0.54 cm), C is the protein concentration (mg/mL), and ε is the absorption coefficient 22,000 M^−1^cm^−1^.

#### 2.3.6. Particle Size

Particle size was determined using a previously established method with minor adjustments [33]. Proteins were standardized to a concentration of 2 mg/mL with 50 mM PBS (pH 6.0, 0.6 M NaCl). Subsequently, the particle size distribution, *D*_3,2_ (meaning diameter in surface) and *D*_4,3_ (meaning diameter in volume) of pork paste proteins were assessed by using a Masterizer 3000 laser particle size analyzer (Malvern Instruments, Malvern, UK). The relative refractive index was set to 1.414, and the absorption rate was 0.001.

#### 2.3.7. SDS-PAGE

In order to observe the cross-linking of paste protein from pork through MAP treatment, reducing and non-reducing gel electrophoresis were performed by referring to the method of Grossi et al. [34] with slight adjustments. Protein samples (2 mg/mL) were combined with an equivalent volume of loading buffer (100 mM Tris, 4% SDS, 20% Glycerol, 1% BPB), with or without 5% β-ME, and subsequently boiled for 5 min at 95 °C. SDS–PAGE was conducted using a 10% separating gel and a 4% acrylamide stacking gel. Standard protein (5 μL) was added to each gel as a reference marker, and 10 μg of protein samples were loaded onto each well. The operating conditions were 90 V for the first 30 min and 120 V for the latter 60 min. After the electrophoresis was completed, the gel was stained with 0.025% Coomassie brilliant blue G250 for 60 min, followed by destaining with a solution composed of 10% methanol and 10% acetic acid. Protein profiles were scanned with a Geno Sens 2100 Gel Imaging System (Qinxiang Scientific Instrument Co., Ltd., Shanghai, China).

### 2.4. Raman Spectra

The secondary structure of the gel protein was detected by using confocal laser Raman spectroscopy. The gel was placed onto the sample cell. The detection parameters were set as follows: the laser wavelength was 532 nm with a power of 20 mW and the number of scans was 30, each with an exposure time of 15 s. Scanning was performed within the range of 700 cm^−1^ to 2000 cm^−1^ at 25 °C and the resolution was 1 cm^−1^. Subsequent to data collection, the curve underwent smoothing, and the multi-point baseline was calibrated using the spectral processing software Labspec version 5.0. The Alix calculation method was used to analyze the secondary structure of the protein.

### 2.5. Chemical Forces

The gel underwent treatments with different specific chemicals for their capacity to cleave distinct bonds. The following solutions were used: 0.6 M NaCl (S1), 0.6 M NaCl + 1.5 M urea (S2), 0.6 M NaCl + 8 M urea (S3), and 0.6 M NaCl + 8 M urea + 0.5 M β-ME (S4). The procedures were conducted by following the description of Wu et al. [35]. S1, S2, S3, and S4 correspond to ionic bond, hydrogen bond, hydrophobic interaction, and disulfide bond, respectively, which were expressed as the percentage of soluble protein in total protein.

### 2.6. Low-Field NMR

The water distribution of the gel was measured using a NiumagPQ001 low-field nuclear magnetic resonance coimaging analyzer (Newmai Electronic Technology Co., Ltd., Shanghai, China) to determine the transverse relaxation time (*T*_2_). After the gel (5 g) stored at 4 °C was recovered to room temperature, it was transferred to the nuclear magnetic tube. The relevant test parameters were set as follows: the resonance frequency was 40 MHz, the interval pulses were 100 μs, and there were 15,000 scanning echoes, and 16 repeated scans. The distribution index fitting analysis of *T*_2_ relaxation data was performed using the Carr-Purcell-Meiboom-Gill (CPMG) sequence, and each sample was repeated six times.

### 2.7. Scanning Electron Microscopy (SEM)

The sample in the shape of 5 mm × 5 mm × 1 mm was cut from the intact gel and then fixed with 2.5% glutaraldehyde for 12 h at 4 °C. The fixed samples were rinsed 3 times using 0.2 M PBS (pH 7.2), then dehydrated with a series of gradient ethanol at one time. After freeze-drying, the gel was sprayed with gold. Finally, the gel was imaged by a Gemini 300 SEM (Zeiss, Oberkochen, Germany) at an acceleration voltage of 5 kV and a magnification of 2000 times [36].

### 2.8. Statistical Analysis

Analysis of variance (ANOVA) for the obtained data was carried out by using the SAS statistical software (Version 9.1.3). The difference of means among treatments was evaluated by Duncan’s multiple range test, and the results were presented as mean ± standard error. The significance threshold was chosen at *p* < 0.05. Principal component analysis (PCA) and Discriminant analysis (OPLS-DA) were performed using SIMCA software (Version 14.1, Sartorius, Göttingen, Germany).

## 3. Results

### 3.1. Color, Texture and Water Holding Capacity

The properties of the gel from minced meat, including color, water holding capacity, and texture are presented in Table 1. The color characteristics of meat products could affect the satisfaction of consumers to a large extent, and the gel whiteness and Δ*E* values were considered to be a useful indicator for the overall color evaluation [24,37]. As shown in Table 1, with the increase of O_2_ concentration (0–80%) in MAP, Δ*E*, whiteness and *L** values of the gel slightly increased, but the difference was not significant (*p* > 0.05). The *a** and *b** of gel decreased with O_2_ concentration from 0% to 20%, and then slightly increased in the latter 40–80% O_2_ treatment. Lower *a** values in MAP groups (0–80%) were observed when compared to Day 0 samples. This may be due to the fact that the O_2_ concentration increases meat oxidation, potentially leading to the unfolding of protein structure, which affects the interaction between protein and water. The increased water absorption within the gel network may contribute to a higher light reflection. That was evidenced by the detection of water holding capacity, among which cooking loss and centrifugation loss of the gel significantly decreased within 0–60% of O_2_ concentration in MAP (*p* < 0.05). However, both cooking loss and centrifugation loss in the 80% O_2_ of MAP group were higher than that of the 60% O_2_ group (Table 1, *p* < 0.05), suggesting reduced water holding capacity by high O_2_ MAP treatment. Similarly, Zhang et al. [26] showed that MP oxidation caused by the increase of hydrogen peroxide (H_2_O_2_, 0–20 mM) concentration initially increased gel water holding capacity and then decreased. Moreover, Zhou et al. [38] reported that the water holding capacity of the gel was enhanced due to the moderate oxidation of the protein, while the gel quality was reduced due to excessive oxidation. Mild oxidation can promote cross-linking between proteins and make the gel network denser and more uniform, thereby reducing the fluidity of water and fixing water in the gel network to improve the water holding capacity of the gel. However, excessive oxidation leads to increased pores in the gel and changes in the water states of the gel, leading to the weak water holding capacity.

Textural properties, including hardness, cohesiveness, springiness, gumminess, and chewiness are essential for assessing the quality of comminuted meat products. The increasing values in hardness, cohesiveness, springiness, gumminess, and chewiness of gel were observed for pork under 0–60% O_2_ of MAP treatment, showing significantly higher values than those of Day 0 pork (*p* < 0.05). However, the textural properties of gel tended to decline when pork was exposed to 80% O_2_ of MAP. Similarly, the instron rupture force of MP gel tended to increase when exposed to oxidizing conditions below 0.5 mM H_2_O_2_ in an iron-catalyzed oxidizing system, but decreased in oxidizing conditions over 1 mM H_2_O_2_ and in metmyoglobin-oxidizing systems, as demonstrated in the study of Xiong et al. [39]. Though proteins were oxidized during incubation in different oxidation systems, intramolecular cross-linking formed to some extent to facilitate gel strength and texture [40]. Otherwise, under strong oxidizing conditions, excessive protein cross-linking produces larger protein aggregates, which may hinder ordered interactions of reactive groups of proteins, while restraining the formation of a compact gel network [41,42].

### 3.2. Oxidation of the Pork Paste

#### 3.2.1. Protein Surface Hydrophobicity and Solubility

The binding of BPB to protein molecules indicates the hydrophobicity of the protein surface and the change in protein conformation [43]. As can be seen in Figure 1a, protein surface hydrophobicity significantly increased over 0–80% O_2_ in MAP treatment groups, which presented higher values than those of Day 0 samples. In addition, the solubility of pork protein in MAP gradually decreased with the increase of O_2_ concentrations (*p* < 0.05). The change in hydrophobicity interaction and structural conformation of proteins was caused by the folding, unfolding, polymerization, and aggregation of proteins under the action of external factors [44]. Protein oxidation has been demonstrated to change the structure of protein and influence protein solubility, which is in accordance with the current study [45,46].

#### 3.2.2. Sulfhydryl Content and Disulfide Bond

The reduction of the active sulfhydryl and total sulfhydryl groups of protein and the formation of disulfide bonds were two important characteristics of protein oxidation. With the increasing O_2_ concentration in MAP, the active sulfhydryl and total sulfhydryl groups in pork paste proteins decreased significantly and the content of disulfide bonds gradually increased (*p* < 0.05), as shown in Figure 1b and Figure 1c, respectively. This discovery was consistent with the study of Delles et al. [47], which suggested that O_2_ induced a pronounced depletion of active sulfhydryl groups and facilitated the development of disulfide cross-links. This occurrence can be attributed to the generation of disulfide bridges at both intra- and inter-molecular levels by the oxidation of protein cysteine residues. Additionally, oxidation may also cause protein unfolding, leading to the exposure of internal protein sulfhydryl groups, which can be counteracted by the concurrent formation of disulfide bonds. Consequently, this dual process resulted in a reduction in the active sulfhydryl content [48].

#### 3.2.3. Dityrosine, Carbonyl Content, and Particle Size

The protein dityrosine and carbonyl content of pork paste are shown in Figure 1d and Figure 1e, respectively. With the increase of O_2_ concentration in MAP, the dityrosine and carbonyl content significantly increased (*p* < 0.05). When protein side chains were exposed to reactive oxygen species (ROS), the generation of tyrosyl radicals occurs. Two tyrosine radicals, whether situated on the same or different protein chains, readily combine to produce dityrosine, which was another pattern of protein cross-linking [49]. Protein carbonyl compounds were reported as the typical products of protein oxidation during meat storage, which affected protein solubility and showed a negative relationship between them [50].

The particle size distribution of protein in pork paste was detected and is presented in Figure 1f. It was observed that the peak of smaller particle size generally increased from the Day 0 sample to the 0–80% O_2_ in MAP samples. Notably, a new peak of higher particles emerged from the 40% O_2_ in the MAP group, and the particle size became higher and larger in the 60% and 80% O_2_ treatment groups. As is shown in Appendix A, the *D*_3,2_ and *D*_4,3_ were significantly increased with O_2_ concentration from 0 to 80% in MAP (*p* < 0.05, Appendix A). It is implied that the average particle size of protein increased with the rise of O_2_ in the MAP of pork. This result may be due to the formation of disulfide bonds and macromolecular aggregates between the protein molecules, as evidenced by Sun et al. [51], finding that protein oxidation is closely related to the changes in protein particle size.

#### 3.2.4. SDS-PAGE of the Paste Protein

The oxidation profiles of paste protein as affected by different O_2_ concentrations in MAP were detected by using reduced gel and non-reduced gel electrophoresis. As seen in Figure 2, with the increase of O_2_ concentration in MAP, the band intensity of protein on the top of the stacking gel strengthened (band 1) and the intensity of protein bands 2, 3, and 4 were visibly weakened. Most changes in protein band intensity in non-reduced gel (band 1–4, Figure 2a) among treatment groups seemed to disappear in reduced gels (band 1′–4′, Figure 2b), suggesting the oxidation of proteins can be largely recovered by the reducing agent. It also showed that the oxidation intensity of MP was exacerbated with the increase of O_2_ concentration, resulting in the aggravation and cross-linking of MP. Consistent with our finding (Figure 1), Li et al. [52] also found that protein intensities of myosin heavy chain and actin were gradually decreased by the treatment of increasing H_2_O_2_ concentration, especially under the condition of high H_2_O_2_ concentration. Moreover, the large aggregates were mainly cross-linked by myosin and proteins in MP through disulfide bonds.

### 3.3. Secondary Structure of the Gel

As indicated in the Raman spectra (Figure 3a), protein secondary structures of the gel were analyzed and the variations were found in groups of Day 0 samples and different O_2_ concentrations in MAP. Predominantly, the information on protein secondary structure was derived from the Amide I band (1600–1700 cm^−1^), primarily involving C-O stretching [53]. The bands from 1650–1658 cm^−1^ correspond to the α-helix, 1665–1680 cm^−1^ correspond to β-sheet, 1680–1690 cm^−1^ correspond to β-turn, and 1660–1665 cm^−1^ correspond to random coil [54,55]. With the increase of O_2_ concentration in MAP, the protein α-helix of gel gradually declined (*p* < 0.05), while β-sheet increased (*p* < 0.05) and minor changes of β-turn and random coil were observed (Figure 3b). This suggests protein structural conversions of gel from the α-helix to other structures, which can possibly be attributed to the O_2_-induced protein oxidation expediting the unfolding of the α-helix. The enhancement of the β-sheet indicates the formation of inter-molecular hydrogen bonding between peptide chains and an ordered rearrangement of protein molecules [56]. In addition, a high ratio of β-sheet was shown to facilitate the assembly of a dense gel network, resulting in improved water retention capacity [57], which is proven by the present study.

### 3.4. Chemical Forces in the Gel

As shown in Figure 4, the intermolecular chemical forces of the gel were determined among treatment groups. It was found that hydrophobic interaction and disulfide bonds occupied a higher proportion in gel compared to ionic and hydrogen bonds, suggesting that hydrophobic interaction and disulfide bonds contributed significantly to the overall chemical forces in all gels, highlighting their dominant role in stabilizing the gel structure. These findings are aligned with previous reports [35,58]. In addition, O_2_ in MAP had minimal impact on ionic and hydrogen bonds during heat-induced gelation while the increasing O_2_ in MAP treatments notably intensified hydrophobic interactions within the gels. The increase of hydrophobic interactions seemingly resulted from the unfolding of protein conformation and the breakdown of protein aggregates by the attack of O_2_ in MAP. Most importantly, the disulfide bond in the gel exhibited a significant rise over 0–60% O_2_ treatment groups but displayed a declining trend in the 80% O_2_ treatment group. Concurrently, the fluctuation in disulfide bonds was similar to the changes in the gel texture (Table 1). This suggested a close connection between the elevated disulfide bonds and the observed alterations in gel strength over 0–80% O_2_ of MAP treatments. Nevertheless, the trend of disulfide bonds in pork paste protein was inconsistent with that in gels, in particular in the 80% O_2_ treatment group, which exhibited higher disulfide bonds in pork paste but a lower value in gel in comparison to that of the 60% O_2_ treatment group. It is known that heating is capable of causing the cleavage of former disulfide bonds, then protein unfolding and the activation of buried sulfhydryl groups into forming new intermolecular disulfide bonds of MPs [59,60]. However, excessive oxidation may lead to a decrease in disulfide bonds and cross-linking between proteins during the process of heat-induced gelation, affecting the structure and texture of the final gel.

### 3.5. Water Distribution and Mobility

As shown in Figure 5, the NMR curves for all gels exhibited three distinct peaks, *T*_21_ (0–5 ms), *T*_22_ (30–200 ms), and *T*_23_ (700–2000 ms), which signified tightly bound water, immobilized water, and free water, respectively [61]. In particular, a shorter relaxation time (*T*_2_) implies reduced water mobility and intensified interaction between water molecules and proteins within the gel structure [62]. As presented in Table 2, with the O_2_ concentration from 0% to 60%, *T*_21_ and *T*_22_ of gel decreased, from 1.18 ms to 0.79 ms, from 79.25 ms to 73.07 ms, respectively. Nevertheless, when O_2_ concentration reached 80%, an increase in *T*_21_ and *T*_22_ was observed, suggesting the water fluidity was raised and flowed out more easily during processing. Moreover, the proportions of peak areas corresponding to the three states of water were shown as *P*_21_, *P*_22_, and *P*_23_. The immobilized water (*P*_22_) occupied the major constituent in the gel (91–95%), which was followed by free water and bound water. The *P*_22_ significantly increased from 92.50% to 94.82%, with an increase in O_2_ concentration (0–60%). Concurrently, *P*_23_ experienced a notable drop from 6.10% to 3.44%. This observation suggests that protein oxidation not only hinders water fluidity in the gel but also promotes the transformation of free water into immobile water, indicating increased water entrapment within the protein structure, leading to immobilization. These were also in accordance with the detection of water holding capacity. According to chemical forces in the gel, changes in inter- and intramolecular bonds within proteins, induced by oxidation, influenced proton accessibility, thereby causing alterations in the relaxation velocity of water in close proximity to protein molecules and its retention ability.

### 3.6. Ultrastructure of Gel

Microstructural analysis was conducted to elucidate variations in the network structure of the gel subjected to MAP treatment at different O_2_ concentrations. As presented in Figure 6, the control gel (Day 0) appeared to be a coarse three-dimensional gel network. With the increase of O_2_ concentration, a compact and homogenous gel network was generated, notably in the group of 60% O_2_, exhibiting gel characteristics of small, uniform pores and filamentous structure. However, in the 80% O_2_ treatment group, the gel structure was tight and irregular, and large aggregates and pore sizes appeared in the microstructure. Nyaisaba et al. [63] stated that the microstructure of heat-induced protein gel was closely associated with the relative velocity of protein unfolding and aggregation. The oxidation of protein, induced by different O_2_ concentrations in MAP can affect its gel properties via structural and conformational changes to the ultrastructural formation of gel during heating.

### 3.7. SDS-PAGE of Gel

The non-participating proteins in the gel formation are presented in Figure 7. It was observed that the thick bands were low molecular weight proteins (<36 kDa). A previous study demonstrated that tropomyosin (a doublet in 34–36 kDa) and troponin (about 20 kDa) had a marginal role in gel formation [64]. Consistent with the report of Gao et al. [65], myosin heavy chain (MHC) and actin nearly vanished in all gels with reduced form and non-reduced forms, providing additional confirmation of their predominant role in the gelation process. With the increase in O_2_ concentration, the intensity of the bands 2, 3, and 3′ was gradually weakened, but the intensity of the bands 4 and 4′ was strengthened in the respective non-reduced gel (Figure 7a) and reduced gel (Figure 7b). In addition, it should be noted that in reduced gels, a tenuous band 1′ and 2′ vanished with the increase in O_2_ concentration. The findings suggested that MAP treatment could effectively promote more proteins to become involved in the construction of gel network structure.

### 3.8. Dependencies between Protein Oxidation and Gel Properties

The linkage of protein oxidation and gel properties of minced pork in the Day 0 sample and MAP samples with different O_2_ concentrations were holistically investigated using principal component analysis (PCA) and discriminant analysis (OPLS-DA) (Figure 8). PCA analysis using an unsupervised technique was used to explore the similarities and hidden patterns between different groups [66]. It could be seen from Figure 8a that the data of various indicators in different processing groups had good repeatability and could be clearly distinguished. The data results of different treatment groups were clearly classified according to the first two principal components (PC1 was 52.8%, PC2 was 16.8%), covering 69.6% of the total data variance. The separable points on the PCA score plot indicate that there were significant differences in the degree of protein oxidation and gel properties of the samples after MAP at different O_2_ concentrations (Figure 8b). In order to better classify and interpret the protein oxidation degree and gel properties of different O_2_ concentration MAP treatments, as presented in Figure 8c, OPLS-DA analysis successfully isolated the significant differences between different O_2_ concentrations in MAP treatment groups. The variable importance in projection (VIP) score in Figure 8d assessed the most important indicators of protein oxidation degree and gel properties affecting different treatment groups, and identified a total of 11 indicators with significant contributions of variables (protein solubility, particle size, α-Helix, β-Sheet, dityrosine, active sulfhydryls, *T*_22_, disulfide bond, chewiness, centrifugation loss, and β-Turn). Therefore, it can be seen from Figure 8 that oxidation affected the properties of pork paste proteins and gels. There is a certain internal relationship between protein oxidation and gel properties.

In summary, the mechanism of oxidation-induced changes of gel properties by O_2_ in MAP is illustrated in Figure 9. It is indicated that as the O_2_ concentration increases, the oxidation degree of MP intensifies, leading to a more intricate protein cross-linking. During the protein gel formation induced by heat treatment, the transition of protein molecules from their native state to a denatured state encompasses alterations in secondary and tertiary structure conformations. This process involves various chemical forces such as ionic bonds, hydrogen bonds, hydrophobic interactions, and disulfide bonds. These modifications ultimately govern the texture and structure of protein gels. Oxidation modification of the protein indirectly influences the gel quality. As the O_2_ concentration rises, the three-dimensional network structure of the gel becomes more uniform, enhancing both texture and water retention. However, when the O_2_ concentration reaches 80%, the protein aggregates into larger structures, thereby diminishing the distinctive characteristics and microstructure of the gel. Thus, our results show that the mechanism of oxidation-induced enhancement of gel properties is through oxidative modification of protein structure, which aggravates protein cross-linking, but excessive oxidative modification will lead to deterioration of the gel, excessive denaturation, and irregular aggregation of internal molecules.

## 4. Conclusions

The primary outcome of this study was that moderate oxidation in MAP (~60% O_2_) markedly improved the gelling performance of paste protein during thermally-induced gelation, as corroborated by the enhanced gel texture and water retention capacity. Moreover, *T*_2_ relaxation analysis demonstrated that moderate oxidation can decrease the water fluidity during the gelation of paste protein and prompted the transformation of free water into immobilized water. The oxidation enhanced protein-protein interactions and triggered a more compact and homogenous gel network. Gel properties, chemical forces, and the ultrastructure of gels were shown to differ significantly between packaging with different O_2_ concentrations. In the case of 60% O_2_ in MAP, the gel had greater textural properties, stronger elasticity, and a denser structure. Collectively, the present study would be beneficial for the application of MAP as a potential tool to regulate meat proteins with improved gel performance via oxidative modification.

## Figures and Tables

**Figure 1 foods-13-00391-f001:**
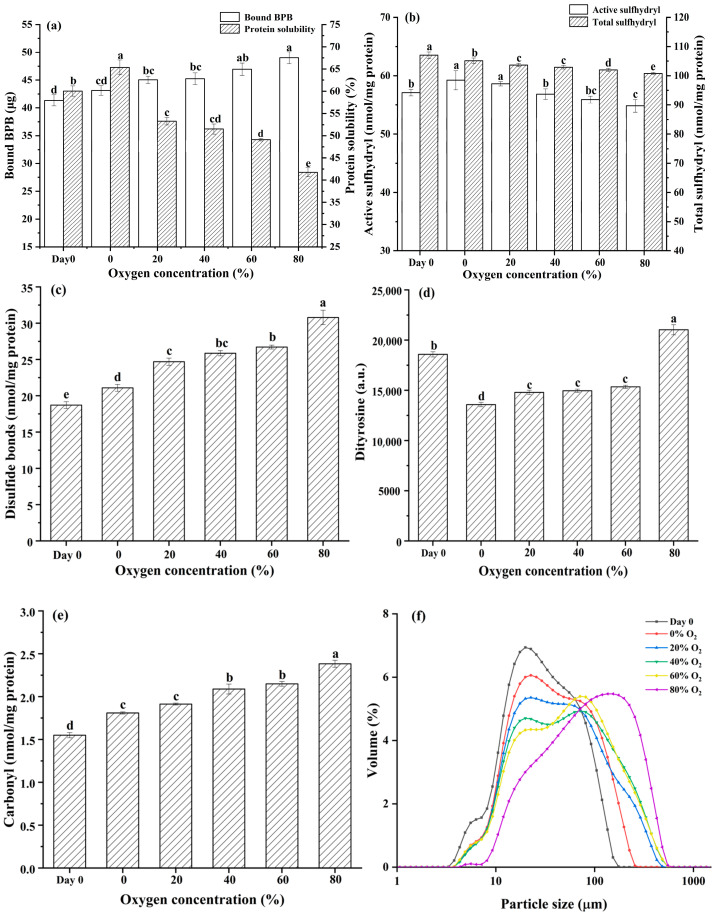
Surface hydrophobicity and protein solubility (**a**), active sulfhydryl and total sulfhydryl groups (**b**), disulfide bonds (**c**), dityrosine contents (**d**), carbonyl contents (**e**), and particle size distribution (**f**) of paste protein from pork before storage (Day 0) and after being placed in refrigerated storage for 5 days in a modified atmosphere with different O_2_ concentrations. Note: the bar indicates standard error of means. Different lowercase letters indicate significant differences among treatment groups (*p* < 0.05).

**Figure 2 foods-13-00391-f002:**
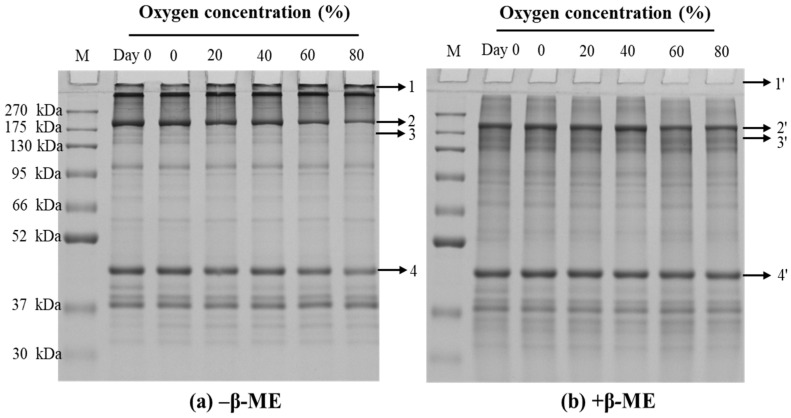
Representative SDS–PAGE patterns of paste protein in non-reduced form ((**a**) −β-ME) and reduced form ((**b**) +β-ME) from pork before storage (Day 0) and after being placed refrigerated storage for 5 days in a modified atmosphere with different O_2_ concentrations.

**Figure 3 foods-13-00391-f003:**
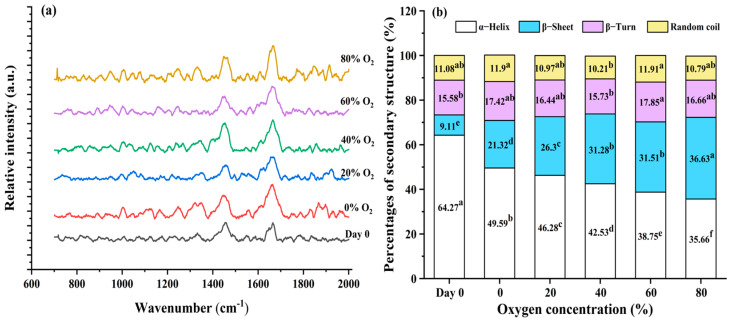
Raman spectra (**a**) and secondary structures (**b**) of pork paste gels from minced meat before storage (Day 0) and after being placed in refrigerated storage for 5 days in a modified atmosphere with different O_2_ concentrations. Note: different lowercase letters indicate significant differences among treatment groups (*p* < 0.05).

**Figure 4 foods-13-00391-f004:**
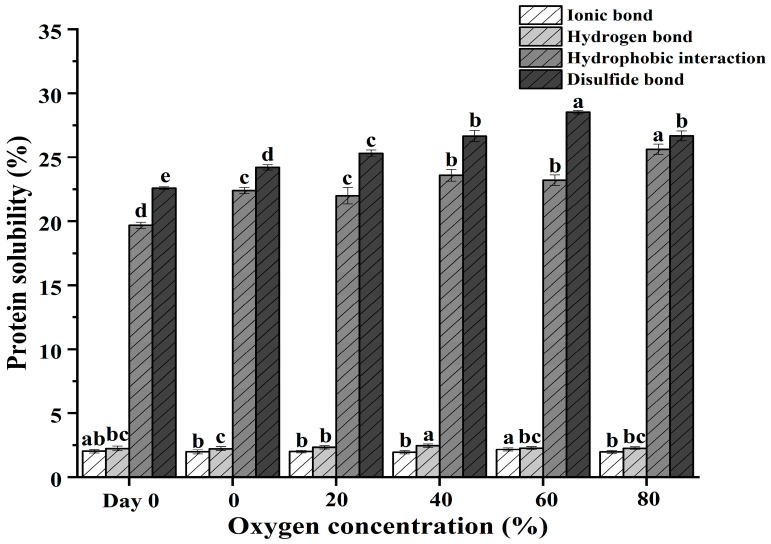
Chemical forces of pork paste gels from meat before storage (Day 0) and after being placed in refrigerated storage for 5 days in a modified atmosphere with different O_2_ concentrations. Note: the bar indicates standard error of means. Different lowercase letters indicate significant differences among treatment groups (*p* < 0.05).

**Figure 5 foods-13-00391-f005:**
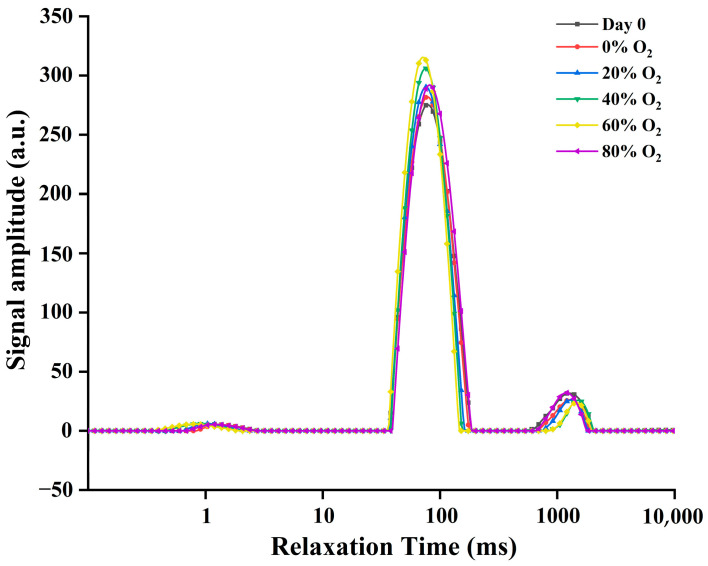
Water distribution of pork paste gels from minced meat before storage (Day 0) and after bring placed in refrigerated storage for 5 days in a modified atmosphere with different O_2_ concentrations.

**Figure 6 foods-13-00391-f006:**
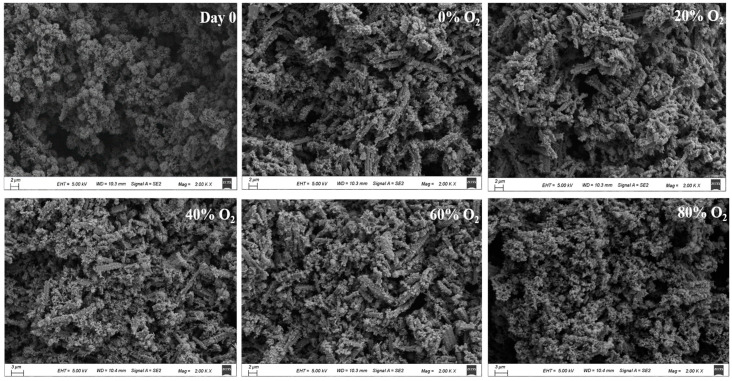
Microstructure of pork paste gels from minced meat before storage (Day 0) and after being placed in refrigerated storage for 5 days in a modified atmosphere with different O_2_ concentrations.

**Figure 7 foods-13-00391-f007:**
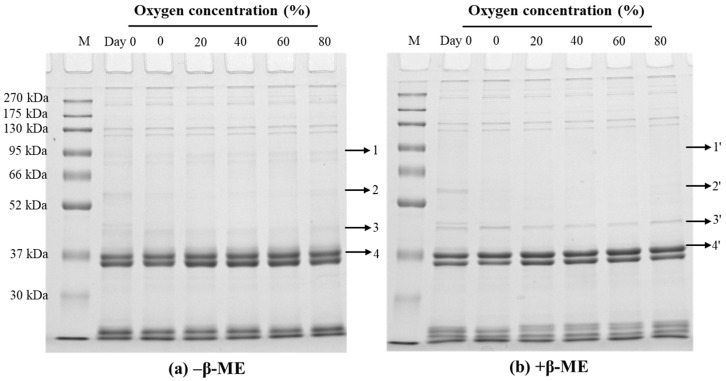
Representative SDS–PAGE patterns of non-participating proteins of pork paste gels from minced meat before storage (Day 0) and after being placed in refrigerated storage for 5 days in a modified atmosphere with different O_2_ concentrations. Samples were prepared in the absence ((**a**) −β-ME) or presence ((**b**) +β-ME) of 5% β-ME.

**Figure 8 foods-13-00391-f008:**
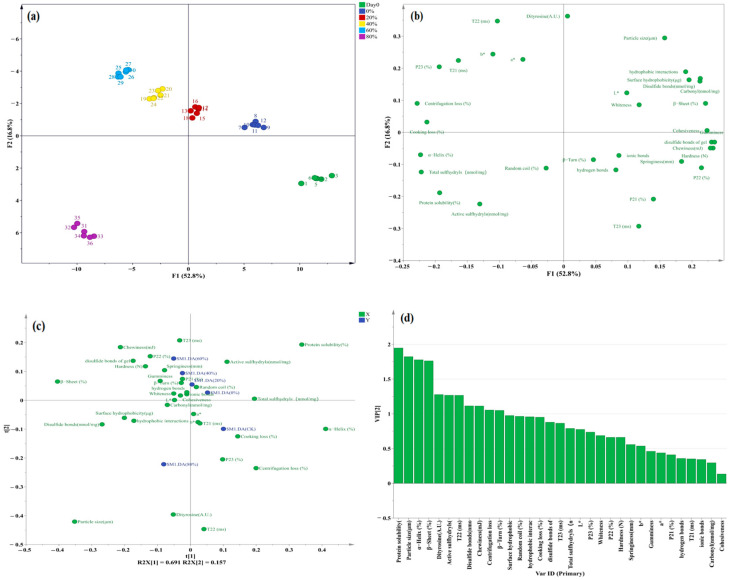
The PCA analysis (score scatter plot (**a**) and loading scatter plot (**b**)) and OPLS-DA analysis (loading scatter plot (**c**) and VIP values (**d**)) of various indexes of paste and gel from minced pork before storage (Day 0) and after being placed in refrigerated storage for 5 days in a modified atmosphere with different O_2_ concentrations.

**Figure 9 foods-13-00391-f009:**
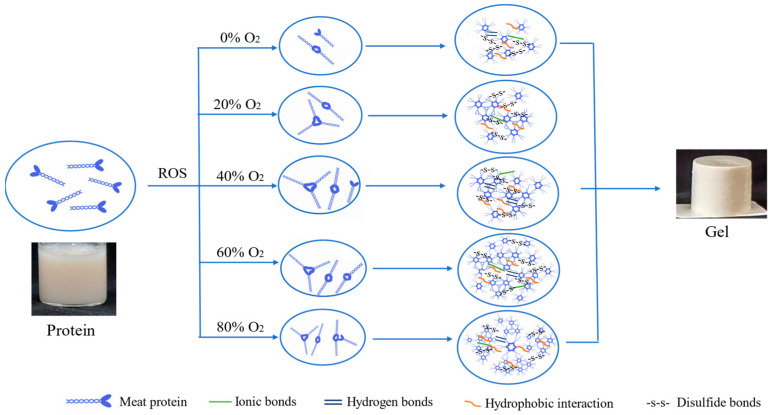
The mechanism diagram of pork paste gels from minced meat before storage (Day 0) and after being placed in refrigerated storage for 5 days in a modified atmosphere with different O_2_ concentrations.

**Table 1 foods-13-00391-t001:** Color, texture, and water holding capacity (cooking loss and centrifugation loss) of pork paste gels from minced meat before storage (Day 0) and after being placed in refrigerated storage for 5 days in MAP with different O_2_ concentrations.

Parameter	Day 0	0%	20%	40%	60%	80%	*p*-Value
Brightness (*L**)	86.83 ± 0.51 ^a^	86.91 ± 0.41 ^a^	86.69 ± 0.47 ^a^	86.69 ± 0.52 ^a^	88.09 ± 0.40 ^a^	87.57 ± 0.52 ^a^	0.274
Redness (*a**)	−0.77 ± 0.01 ^a^	−0.98 ± 0.01 ^bc^	−1.06 ± 0.04 ^c^	−0.87 ± 0.03 ^ab^	−0.98 ± 0.04 ^bc^	−0.86 ± 0.06 ^a^	0.002
Yellowness (*b**)	8.19 ± 0.06 ^a^	8.01 ± 0.05 ^abc^	7.68 ± 0.02 ^c^	7.84 ± 0.08 ^bc^	7.78 ± 0.11 ^bc^	8.05 ± 0.18 ^ab^	0.028
Color differences (Δ*E*)	-	0.60 ± 0.18 ^a^	0.88 ± 0.14 ^a^	0.77 ± 0.24 ^a^	1.24 ± 0.20 ^a^	1.04 ± 0.22 ^a^	0.262
Whiteness	84.47 ± 0.45 ^a^	84.62 ± 0.33 ^a^	84.60 ± 0.42 ^a^	84.52 ± 0.42 ^a^	85.74 ± 0.28 ^a^	85.15 ± 0.35 ^a^	0.204
Hardness (N)	1.71 ± 0.09 ^e^	2.32 ± 0.11 ^d^	3.70 ± 0.10 ^c^	4.50 ± 0.23 ^b^	5.23 ± 0.12 ^a^	4.13 ± 0.20 ^bc^	<0.001
Cohesiveness	0.23 ± 0.01 ^c^	0.29 ± 0.01 ^b^	0.35 ± 0.02 ^a^	0.39 ± 0.02 ^a^	0.39 ± 0.04 ^a^	0.38 ± 0.01 ^a^	<0.001
Springiness (mm)	3.69 ± 0.16 ^b^	4.92 ± 0.28 ^a^	5.20 ± 0.18 ^a^	5.47 ± 0.14 ^a^	5.46 ± 0.10 ^a^	5.21 ± 0.12 ^a^	<0.001
Gumminess	0.50 ± 0.02 ^d^	0.67 ± 0.05 ^d^	1.26 ± 0.04 ^c^	1.77 ± 0.17 ^ab^	2.01 ± 0.05 ^a^	1.57 ± 0.10 ^b^	<0.001
Chewiness (mJ)	2.18 ± 0.17 ^d^	3.30 ± 0.34 ^d^	6.58 ± 0.33 ^c^	9.73 ± 1.04 ^ab^	11.02 ± 0.47 ^a^	8.04 ± 0.54 ^bc^	<0.001
Cooking loss (%)	36.78 ± 0.11 ^a^	35.89 ± 0.22 ^a^	34.35 ± 0.35 ^b^	34.61 ± 0.40 ^b^	31.83 ± 0.41 ^c^	33.84 ± 0.25 ^b^	<0.001
Centrifugation loss (%)	33.76 ± 0.50 ^a^	30.67 ± 0.36 ^b^	29.20 ± 0.36 ^c^	26.77 ± 0.15 ^d^	24.84 ± 0.30 ^e^	28.24 ± 0.25 ^c^	<0.001

Note: Mean values ± standard error. Different lowercase letters (^a–e^) in the same line indicated significant differences among treatment groups at *p* < 0.05.

**Table 2 foods-13-00391-t002:** Corresponding relaxation time (*T*_2_) and peak area proportion (*P*) of pork paste gels from minced meat before storage (Day 0) and after being placed in refrigerated storage for 5 days in a modified atmosphere with different O_2_ concentrations.

Parameter	Day 0	0%	20%	40%	60%	80%	*p* Value
*T*_21_ (ms)	1.18 ± 0.03 ^b^	1.30 ± 0.01 ^a^	1.07 ± 0.02 ^c^	0.88 ± 0.01 ^d^	0.79 ± 0.01 ^e^	1.19 ± 0.00 ^b^	<0.001
*T*_22_ (ms)	79.25 ± 0.40 ^b^	78.23 ± 0.14 ^b^	75.76 ± 0.32 ^c^	74.61 ± 0.75 ^c^	73.07 ± 0.67 ^d^	81.38 ± 0.40 ^a^	<0.001
*T*_23_ (ms)	1256 ± 30.5 ^b^	1275 ± 23.2 ^b^	1312 ± 41.4 ^b^	1438 ± 30.9 ^a^	1420 ± 34.5 ^a^	1217 ± 24.1 ^b^	0.001
*P*_21_ (%)	1.52 ± 0.02 ^b^	1.29 ± 0.06 ^c^	1.55 ± 0.05 ^b^	1.76 ± 0.04 ^a^	1.88 ± 0.05 ^a^	1.43 ± 0.02 ^b^	<0.001
*P*_22_ (%)	91.30 ± 0.25 ^d^	92.50 ± 0.45 ^c^	93.57 ± 0.19 ^b^	94.15 ± 0.08 ^ab^	94.82 ± 0.15 ^a^	93.50 ± 0.46 ^b^	<0.001
*P*_23_ (%)	6.84 ± 0.33 ^a^	6.10 ± 0.39 ^ab^	4.55 ± 0.24 ^c^	4.29 ± 0.15 ^cd^	3.44 ± 0.23 ^d^	5.58 ± 0.50 ^b^	<0.001

Note: Mean values ± standard error. Different lowercase letters (^a–e^) in the same line indicate significant differences between treatment groups at *p* < 0.05.

## Data Availability

Data is contained within the article.

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
