# Peer review of "Oxidative Modification, Structural Conformation, and Gel Properties of Pork Paste Protein Mediated by Oxygen Concentration in Modified Atmosphere Packaging"

_foods, 2024, doi:10.3390/foods13030391_

Round 1

Reviewer 1 Report

Comments and Suggestions for Authors

The topic is interesting. However, some revisions are made before acceptance for the publication in Foods. The detailed information can be found in the attachments.

Author Response

Thank you very much for your comments and suggestions. Please refer to the appendix for details of the amendments. Please refer to the attachment.

Reviewer 2 Report

Comments and Suggestions for Authors

I found the manuscript submitted for review very interesting. The research was conducted with a selection of appropriate research methods, which were described in detail. I consider the research carried out by the authors to be very ambitious and thorough. I have no reservations about them. In my opinion, one drawback is the very small number of animals sampled for the study. However, I am aware that the research was very tedious and costly. I suggest that the authors make it clear that this is a small study group.

The conclusions formulated by the Authors respond to the purpose of the research.

Overall, I evaluate the manuscript very positively and appreciate its applied value.

Author Response

(The authors gave the same response as above.)

Reviewer 3 Report

Comments and Suggestions for Authors

I would like to thank the authors for an interesting paper. You certainly did a lot of work in the lab.

I would recommend some revisions before publication:

Generally, check the verb tense throughout the paper and use what is appropriate for each section. I especially noticed that, in some places, you have used past tense where it should be present tense.

Just to be clear, is paste = gel in your paper? You employ both terms, and I was reading it as being the same thing based on 2.1.2, but then in 3.4 L418-421 it seems like they are two different things.

Abstract

L14 Parentheses: Move O2 to the end, i.e. (0, 20, 40, 60, and 80% O2).

L25 Should be “…capable of reinforcing…”

Introduction

L73 Is there an extra “pork” in this sentence? It is a bit confusing.

Materials and methods

L99 Please, state the piece of equipment used to measure the gas composition in the packages.

L100-102 Please, rewrite this sentence to make the meaning clearer.

L119 Why illuminant C? That is very unconventional for measuring the color of meat, where illuminant D65 or Illuminant A are usually used. Was the observer angle 2° or 10°?

L122 What is the source of this formula? Often, one would calculate the total color difference (∆E) for meat/meat products. I noticed that reference [34], which uses the Whiteness index, deals with squid, and from that paper, it seems like this index is primarily relevant to seafood.

L146 “centrifuged” instead of “centrifugal”

L207ff Perhaps add a line describing the purpose of doing the SDS-PAGE analysis.

L231 What is β-ME?

Results and discussion

L267 If not added to the Materials and Methods section, I think you should explain why whiteness is a relevant parameter to measure for these pork pastes.

L370-372 Please, correct the sentence to make it understandable.

L422 Should be “..is capable of causing…”

L480 The abbreviation MCH is not explained.

L510 VIP score?

Conclusion

L544 Are you sure that “thermal-induced gelation” shouldn’t instead be “thermally induced gelation”?

Comments on the Quality of English Language

Generally, the paper is understandable, but some corrections are needed (see general comments above).

Author Response

(The authors gave the same response as above.)

Reviewer 4 Report

Comments and Suggestions for Authors

This study is original well organized and well presented. Authors have provide various techniqs to support their finding and are combining very good their results. A;thought some more attention must be given in some additional points. Please fsee the attached file

Author Response

(The authors gave the same response as above.)

Round 2

Reviewer 1 Report

Comments and Suggestions for Authors

I am satisfied with the author's response and revision. Therefore, I am recommending it for publication.